# Pharmacokinetics and Pharmacodynamics Modeling and Simulation Systems to Support the Development and Regulation of Liposomal Drugs

**DOI:** 10.3390/pharmaceutics11030110

**Published:** 2019-03-07

**Authors:** Hua He, Dongfen Yuan, Yun Wu, Yanguang Cao

**Affiliations:** 1Center of Drug Metabolism and Pharmacokinetics, China Pharmaceutical University, Nanjing 210009, China; huahe827@163.com; 2Division of Pharmacotherapy and Experimental Therapeutics, School of Pharmacy, University of North Carolina at Chapel Hill, Chapel Hill, NC 27599, USA; dyuan@email.unc.edu; 3Department of Biomedical Engineering, University at Buffalo, The State University of New York, 332 Bonner Hall, Buffalo, NY 14260, USA; ywu32@buffalo.edu; 4Lineberger Comprehensive Cancer Center, University of North Carolina at Chapel Hill, Chapel Hill, NC 27599, USA

**Keywords:** pharmacokinetic–pharmacodynamics, modeling and simulation, PBPK model, liposomal drugs, EPR effect, regulatory review

## Abstract

Liposomal formulations have been developed to improve the therapeutic index of encapsulated drugs by altering the balance of on- and off-targeted distribution. The improved therapeutic efficacy of liposomal drugs is primarily attributed to enhanced distribution at the sites of action. The targeted distribution of liposomal drugs depends not only on the physicochemical properties of the liposomes, but also on multiple components of the biological system. Pharmacokinetic–pharmacodynamic (PK–PD) modeling has recently emerged as a useful tool with which to assess the impact of formulation- and system-specific factors on the targeted disposition and therapeutic efficacy of liposomal drugs. The use of PK–PD modeling to facilitate the development and regulatory reviews of generic versions of liposomal drugs recently drew the attention of the U.S. Food and Drug Administration. The present review summarizes the physiological factors that affect the targeted delivery of liposomal drugs, challenges that influence the development and regulation of liposomal drugs, and the application of PK–PD modeling and simulation systems to address these challenges.

## 1. Introduction

Liposomes were first described by British hematologist, Alec Bangham, in 1961 [1]. Shortly thereafter, the concept of liposomal drug delivery systems rapidly gained popularity [2,3,4]. Liposomes are spherical vesicles that comprise one or more phospholipid bilayers. They are used to improve the solubility, bioavailability, systemic retention, or tissue-related distribution of encapsulated drugs [3,5,6].

By increasing on-target drug accumulation and lowering off-target drug distribution, liposome-based delivery systems have the ability to significantly widen the therapeutic window. In particular, liposomes have been demonstrated to constitute an effective drug delivery system in oncology as they have the capacity to selectively deliver cytotoxic drugs to solid tumors [7,8,9,10]. Maeda et al. suggested that enhanced permeability and retention (EPR) were fundamental to the increased delivery of drugs by liposomes to the tumor [11]. The EPR effect is based on observations that tumor tissues often have an abnormal tumor vasculature, an aberrant vascular architecture, and a lack of lymphatic drainage [12]. In 1995, the first liposomal drug, Doxil^®^, was approved by the U.S. Food and Drug Administration (FDA) to treat ovarian cancer and acquired immune deficiency syndrome-related Kaposi’s sarcoma [13]. Investment in the development of liposomal drugs has grown exponentially over the past two decades. A number of strategies have been proposed to further optimize liposome delivery systems [14]. By 2018, a total of 16 liposomal drugs had been approved by the FDA and/or European Medicines Agency (EMA).

The physicochemical properties of liposomes, including size, surface charge, shape, and surface coating, can be potentially engineered to improve the targeted delivery and enhance the pharmacological efficacy of encapsulated drugs, making liposomes a versatile delivery system [15]. The multifunctionality of liposomes provides a broad design space in which to improve drug delivery efficiency. However, it often results in intricate interactions with multiple components of the biological system, leading to varying in vivo disposition behaviors, when compared to their small molecule counterparts. Considerable effort has been made in trying to establish a quantitative relationship between the physicochemical properties of liposomes and related in vivo disposition behavior. Nevertheless, the ability to accurately predict the disposition characteristics and biological interactions of liposomes based on their physicochemical properties has generally remained unattainable. The dynamic nature of the biological system and the multiple biological components that are involved in interactions with liposomes makes it challenging to predict the in vivo disposition of liposomal drugs.

Pharmacokinetic–pharmacodynamic (PK–PD) modeling and simulation is widely applied in drug development and regulation systems. Applications include the translation of preclinical observations to the clinical setting, the selection of dosing regimens in special populations, and the optimization of clinical trials [16]. PK–PD modeling and simulation systems are a systems-based approach that enables the joint analysis of system- and drug-specific factors and has shown significant potential in identifying the multiscale mechanisms of liposomal disposition, quantifying and optimizing delivery systems, and correlating the physicochemical properties with systemic performance [17,18].

The current developmental landscape of liposomal drugs, with a special focus on PK–PD properties, is evaluated in this paper. The biophysical barriers to targeted drug delivery using liposomes are reviewed, in keeping with the challenges associated with liposomal drug development and the regulatory assessment. In particular, PK–PD modeling and simulation systems are investigated as a way of addressing these challenges.

## 2. Liposomal Drugs Approved for Use in Humans

Since the first liposomal drug, Doxil^®^, was endorsed in 1995, 16 liposomal drugs have been approved for human use. Most liposomal drugs are indicated for cancer therapy (Table 1), as per the EPR hypothesis. Many liposomes have also been developed for the delivery of anti-infective drugs, vaccines, analgesics, and photodynamic therapeutic agents. Bulbake et al. listed the liposomes that have been approved for use in humans in their review [14]. The PK–PD profiles of the approved liposomal drugs were the focus of the current research.

### 2.1. Liposomal Drugs for Cancer Treatment

Doxorubicin is one of the most effective anti-cancer drugs used to treat a broad range of cancers [19]. However, accumulative dose-related cardiotoxicity drastically restricts its clinical application [20,21]. Thus, liposomes were developed to improve the therapeutic index of doxorubicin by selectively delivering it to tumors, while reducing its accumulation in the heart. Currently, two types of liposomal doxorubicin, i.e., Doxil^®^ (or Caelyx^®^ in Europe) and Myocet^®^, have been approved for clinical use. The surface modification of liposomes with polyethylene glycol (PEG) is the major difference between them [22]. Doxil^®^ (Caelyx^®^), pegylated liposomal doxorubicin, was the first liposomal drug approved by the FDA. Myocet^®^, non-pegylated liposomal doxorubicin, is currently licensed in Europe and Canada. The encapsulation of doxorubicin in these two types of liposomes remarkably changes the PK profile and tissue distribution of doxorubicin. In studies on humans, the volume of distribution decreased from 254 L to 142 L or to 4 L, and systemic clearance was reduced from 45 L/hour to 9.7 L/hour or to 0.1 L/hour for Myocet^®^ and Doxil, respectively [23,24]. Liposomal doxorubicin accumulation in the targeted tumors was reported to be 4~16-fold higher than free doxorubicin accumulation [23]. Importantly, both Doxil^®^ and Myocet^®^ substantially reduced cardiotoxicity [25,26,27]. Despite the improvement in disposition profiles, only a moderate increase in patient survival benefit was seen with the use of both liposomal delivery systems for doxorubicin, suggesting that a large gap remains in the translation of targeted delivery to improved clinical outcomes [26,27].

Apart from Doxil^®^ and Myocet^®^, Lipodox^®^, a generic form of Doxil^®^, gained expedited approval in 2012 in response to a drug shortage of Doxil^®^ in the United States. However, by early 2019, Lipodox^®^ had not yet been approved by the EMA. PK equivalence (between Doxil^®^ and Lipodox^®^) was evaluated in two clinical studies. Bioequivalence was concluded for total and encapsulated doxorubicin in both trials. However, bioequivalence was only concluded in one trial for free doxorubicin and the other trial concluded that there was nonequivalence due to the small sample size [28]. The plasma drug concentration, a conventional surrogate for small molecules in bioequivalence studies, does not necessarily reflect the bioavailability of the liposomal drug in the intended targeted organ, nor the therapeutic benefit of generic liposomes [29]. The therapeutic equivalence of LipoDox^®^ and Doxil^®^ continues to be debated. Both equivalence and nonequivalence data have been reported in animal models [30,31,32].

Daunorubicin is an anthracycline compound that is primarily used to treat acute myeloid leukemia. Liposomal daunorubicin (DaunoXome^®^) was approved by the FDA in 1996 to treat human immunodeficiency virus-associated Kaposi’s sarcoma [33]. Liposomal daunorubicin was shown to increase daunorubicin accumulation in the tumor by approximately 10-fold in mice [34]. In humans, liposomal daunorubicin was found to significantly prolong systemic retention and increase the targeted exposure of liposomal daunorubicin in comparison to the free drug [35,36].

The combination of cytarabine/daunorubicin at a fixed molar ratio of 5:1 was optimized to exert a synergistic effect in killing the P388 leukemic cell line in vitro [37]. Vyxeos^®^, a liposomal drug, was designed to simultaneously deliver cytarabine/daunorubicin at a fixed molar ratio of 5:1 [37,38]. Vyxeos^®^ was approved in 2017 by the FDA to treat newly diagnosed therapy-related acute myeloid leukemia (AML) or AML with myelodysplasia-related changes in adults [39]. In a preclinical study, Vyxeos^®^ was demonstrated to selectively increase the exposure of daunorubicin and cytarabine in a fixed ratio in leukemia-laden bone marrow, thus resulting in a significantly prolonged effect with regard to leukemia suppression [40]. In clinical trials, it was shown to have superior efficacy and comparable toxicity to the standard combination of daunorubicin and cytarabine in patients with AML [41].

Both vincristine and cytarabine are cell cycle phase-specific chemotherapeutic agents. Prolonged exposure to tumors above a threshold of effective concentration is critical to ensure their therapeutic efficacy. Free forms of vincristine and cytarabine are hardly able to maintain the therapeutic concentration owing to rapid clearance [42,43]. Thus, prolonging systemic retention is a strategy that can be used to increase their efficacy.

DepoCyt^®^ is a liposomal cytarabine that was developed using DepoFoam^™^ technology [44]. DepoCyt^®^ was approved by the FDA in 1999 to treat lymphomatous meningitis. The slow and continuous release profile of liposome resulted in extended (i.e., up to 14 days) and enhanced tumor exposure to cytarabine, thus increasing the response rate when compared with the free cytarabine treatment [45,46,47,48]. Marqibo^®^ is a liposomal vincristine injection that was approved to treat adults with Philadelphia chromosome-negative acute lymphoblastic leukemia [48]. It was designed to gradually release vincristine in the targeted tumors and prolong exposure, with the objective of increasing therapeutic efficacy [49]. Compared to its free counterpart, liposomal vincristine was shown to have lower dose-limiting toxicity and the ability to be administered at higher doses [50]. Marqibo^®^ demonstrated promising antitumor activity in adult patients with refractory or relapsed acute lymphoblastic leukemia in a phase II study, thus ensuring its accelerated approval in 2012 [51].

Muramyl tripeptide phosphatidylethanolamine (MTP-PE), an immunotherapeutic agent, is known to activate macrophages and monocytes in the mononuclear phagocyte system (MPS), including in the liver, spleen, and lungs [52]. Mepact^®^, liposomal MTP-PE, was approved by the EMA as an “orphan” drug in 2004, for use in combination with postoperative multiagent chemotherapy to treat high-grade, resectable, nonmetastatic bone tumors. Liposome-encapsulated MTP-PE was selectively delivered to the MPS. It was shown to be retained in the targeted organs for a long duration [53,54]. In addition to having an improved biodistribution profile, MTP-PE demonstrated high therapeutic efficacy and low off-target toxicity [55,56].

Onivyde^™^ is an irinotecan liposomal injection that is used to treat patients with metastatic pancreatic adenocarcinoma. Liposomal irinotecan was associated with delayed clearance and had a prolonged half-life and reduced volume of distribution in a phase I clinical trial [57]. Compared to free irinotecan, liposomal irinotecan reduced the plasma peak concentration, had a prolonged terminal half-life, and increased the plasma exposure of SN-38, a bioactive metabolite of irinotecan [57]. It was approved in 2015 by the FDA for intended use in combination with fluorouracil and folinic acid, based in part on its prolonged progression-free and overall survival in the nanoliposomal irinotecan with fluorouracil and folinic acid in metastatic pancreatic cancer after previous gemcitabine-based therapy (NAPOLI-1) study [58].

### 2.2. Liposomal Drugs for Other Indications

Initially, amphotericin B was a widely used antifungal agent in the management of severe systemic fungal infections. However, it was discontinued primarily as a result of dose-limiting toxicity, including nephrotoxicity and infusion-related reactions [59,60]. Liposomal amphotericin B (Ambisome^®^) was developed in 1997 to treat a broad range of fungal infections. Amphotericin B is encapsulated in the liposome bilayer and is only released when the liposome binds to the fungus [60]. This liposome formulation significantly improves the safety profile owing to the relatively low concentration of amphotericin B that is released into the blood circulation [61].

A liposomal amikacin inhalation suspension (Arikayce^®^) was recently approved to treat *Mycobacterium avium complex* lung disease. When treating lung infections, it is difficult to deliver an adequate amount of antibiotic to the lung while maintaining nontoxic plasma concentrations. Arikayce^®^ was designed to overcome this challenge via the delivery of a high liposomal antibiotic load to the lungs after nebulization [62]. Compared to intravenously delivered free amikacin, Arikayce^®^ was shown to respectively increase exposure to amikacin by 42-, 69-, and 274-fold in the lung tissues, airways, and macrophages [62]. The drug concentration of Arikayce^®^ in the pulmonary macrophages increased by 5~8-fold, when compared to inhaled free amikacin [62].

Epaxal^®^ is a virosome-based vaccine that is used to ensure the prevention of hepatitis A [63]. Deactivated hepatitis A is attached to the virosome surface. The Epaxal^®^ structure promotes the delivery of the hepatitis A virus antigen to the immunocompetent cells. Compared to conventional aluminum-adsorbed vaccines, Epaxal^®^ confers higher tolerability and fewer adverse effects.

Inflexal^®^ V is a virosomal-adjuvanted, inactivated influenza vaccine. The hemagglutinin is extracted from the influenza virus and is incorporated into the liposomal membranes. In one study, Inflexal^®^ V was shown to have a superior immune response compared to other vaccines, such as commercially available whole-virus and subunit vaccines [64]. Limited PK profiles have been reported for liposomal vaccines.

DepoDur^™^ is an extended-release epidural morphine that was approved in 2004 to manage postsurgical pain. Morphine is encapsulated in lipid foam using DepoFoam^™^ technology [65] to allow its extended release into the epidural space. The maximal morphine concentration in the serum and cerebrospinal fluid was found to be 6% and 32% of that for a dose of morphine sulfate, respectively, after the epidural administration of a single dose of DepoDur^™^ [66]. The pain relief associated with DepoDur^™^ is prolonged, while the supraspinal toxic effect is reduced compared to morphine sulfate, likely due to the sustained release of morphine. Exparel^®^ is a bupivacaine extended-release liposome that was approved in 2011 to treat postsurgical pain. Similar to DepoDur^™^ and DepoCyt^®^, it is also a DepoFoam^™^ technology-based liposome. Compared to the plain formulation, the peak concentrations of bupivacaine for liposomal bupivacaine were comparable (0.87 vs. 0.83 μg/mL) at a 4-fold higher dose, and the time to maximal concentration was 7-fold greater [67], resulting in a lower risk of toxicity. The expanded safe dose range and the delayed elimination of bupivacaine in the liposome formulation produce prolonged analgesia.

Visudyne^®^ is a light-activated drug that was approved to treat patients with age-related macular degeneration-induced neovascularization. Verteporfin was encapsulated in liposomes and utilized as a photosensitizer in photodynamic therapy to selectively eradicate abnormal blood vessels in the eye. The therapeutic benefits of liposomal verteporfin can last for at least 24 months without safety-related challenges when treating age-related macular degeneration [68]. The systemic clearance of verteporfin was not seen to be significantly influenced by the liposome formulation in a biodistribution study [69]. The liposome formulation slightly increased the distribution of verteporfin in the tissues and tumor.

## 3. Physiological Barriers to the Targeted Delivery of Liposomes

The pharmacological effects of chemical compounds are usually limited by their poor physicochemical and PK properties, such as low solubility, suboptimal biodistribution, and rapid clearance. Liposomes are designed to overcome these challenges through the encapsulation of chemical compounds, thereby increasing drug solubility, reducing systemic clearance, and improving drug distribution to the target tissue. However, new challenges have been encountered when seeking to understand and predict the systemic disposition of liposomal formulations. The physiological barriers that limit the delivery of liposomal drugs to the targeted site are evaluated here (Figure 1).

### 3.1. Systemic Clearance

In general, liposomes have the ability to circulate longer than small molecules via a reduction in kidney filtration. However, the in vivo persistence of conventional liposomes was not significantly prolonged as a result of rapid recognition and degradation by the MPS. Once in the bloodstream, the liposomes are quickly opsonized [72] and then recognized and engulfed by the macrophages, leading to their rapid clearance [73]. Specifically, the half-life of doxorubicin was only prolonged from 0.2 h to 2.5 h by Myocet^®^ [74]. Liposome properties, such as surface modification, size, and zeta potential, are all determinants of opsonization [72]. An effective approach to reducing opsonization and prolonging liposomal systemic persistence is to coat the liposome surface with hydrophilic polymers, i.e., PEG [75]. The half-life of liposomal doxorubicin was prolonged from 2.5 h (Myocet^®^) to 55 h (Doxil^®^) using the pegylation technique [74].

### 3.2. Extravasation

Functional blood vessels are essential for the delivery of the drug to the targeted tissues. Unlike the orderly organization of vessels in normal tissues, the arrangement of the tumor vessels is often disorderly, resulting in spatially and temporally heterogeneous blood perfusion [76]. A positive correlation was found in vivo between tumor blood perfusion and the extent of liposome distribution using imaging-based techniques [77,78]. Irregular tumor vasculature is a significant barrier to the targeted delivery of liposomes to the tumor. Vascular endothelial growth factor (VEGF) inhibitors were developed to suppress angiogenesis, so as to reduce the blood supply to the tumor [79]. However, mounting evidence suggests that the clinical benefit of VEGF inhibitors can largely be ascribed to normalization of the chaotic tumor vasculature, resulting in elevated blood perfusion in the tumor [80,81,82]. Thus, VEGF inhibitors are often exploited to improve tumor blood supply and liposome delivery.

Enhanced blood perfusion increases liposome delivery to the tumor, but does not necessarily equate to the increased exposure of the drug to the targeted cells. To reach the action sites, the liposomes need to extravasate first before reaching the targeted cells. The continuous layers of endothelial cells and the surrounding perivascular cells in the normal tissues constitute a physical barrier to the extravasation of the nanoscale liposomes, resulting in their limited distribution in normal tissue. Tumors with a highly abnormal vasculature (e.g., widened inter-endothelial cell gaps, tenuous pericyte cell contacts, and abundant fenestrated capillaries) have higher vessel permeability, thus allowing the passive diffusion of liposomes across the tumor vasculature [83,84,85]. The blood supply, vasculature permeability, particle size, and liposomal surface charge all affect the extravasation of liposomes. Strategies to either reduce the particle size or modify the blood vessels have the potential to enhance the extravasation of liposomes [86]. Over the past three decades, several pharmacological and physical approaches have been developed to ensure blood vessel modification, such as vessel permeabilization, vessel disruption, hyperthermia, and sonoporation [87].

### 3.3. Spatial Penetration in the Tumor Tissues

Physical disruption to the microvascular membrane and collapsed lymphatics leads to the accumulation of plasma proteins in a solid tumor, resulting in interstitial hypertension [88]. Convection is diminished in tumors with elevated interstitial fluid pressure (IFP), and the penetration of liposomes is largely limited to diffusion [89]. In addition, the dense extracellular matrix has been observed to drastically lower the diffusion coefficient of liposomes in tumors [90,91]. Thus, liposomes in tumors are characterized by a concentration gradient that extends from the perivascular area to the distal tumor cells [76,92]. The exposure of drugs to the distal tumor cells is often observed to be several orders of magnitude lower than that in perivascular area [89]. It is believed that insufficient drug exposure compromises therapeutic efficacy. Thus, increasing tumor spatial penetration has emerged as a potentially invaluable approach with which to improve the anti-tumor efficacy of liposomes [89]. For instance, VEGF receptor 2 blockade assists with vascular normalization and has been observed to increase drug penetration by reducing IFP [93]. Ensuring the degradation of the extracellular matrix is another strategy that can be used to increase tumor penetration [94]. Tumor spatial penetration can also be improved by modifying the liposomal properties, i.e., size and surface characteristics [95].

### 3.4. Intracellular Uptake

Delivery of the payloads into the targeted cells is needed to ensure efficacy as most pharmacological targets of the payloads reside intracellularly, particularly for macromolecular agents (e.g., DNA and siRNA). However, a fraction of the liposomes in the interstitial space is degraded before being taken up by the tumor cells due to the acidic environment in the interstitial fluid and rapid phagocytosis by the tumor-associated macrophages. An unstable payload, i.e., a macromolecular agent, is rapidly degraded in the interstitial fluid. This compromises the therapeutic effects. Liposomes enter the tumor cells via various endocytic pathways [96]. Thus, coating them with a targeting antibody can escalate tumor cell uptake [97]. Once the liposomes in the endosome have been taken up by the tumor cells via endocytosis, the liposomes are either recycled to the cell surface or degraded by the acidic conditions or the residing enzymes after the endosome has matured into a late endosome and fused with the lysosome [98,99]. The released payload that escapes from the endosomes binds to the intracellular target to elicit the therapeutic effect. Degradation in the interstitial space and during the endocytosis process acts as a barrier and reduces the amount of payload that reaches the intracellular target.

## 4. Challenges to Liposomal Drug Development and a Regulatory Review

The disposition of the payload (i.e., liposomal drugs) is jointly determined by the kinetics of both the carrier and the payload itself. Compared to conventional formulations, the liposomes interact with multiple components of the physiological system, thereby making it difficult to predict the disposition behaviors of liposomal drugs. The challenges to the development and regulation of liposomal drugs are described in this section.

### 4.1. The Identification of Critical Quantity Attributes

Multiple physicochemical properties (e.g., particle size, zeta potential, lipid phase transition temperature, lipid composition, and drug release kinetics) affect the multiscale mechanisms of liposomal disposition, i.e., systemic clearance, tissue extravasation, tissue penetration, and cellular internalization [100]. The relationship between the physicochemical properties of liposomes and their systemic administration has been extensively investigated, and key guiding principles have been established. Nevertheless, an accurate prediction of liposomal disposition by the physicochemical properties remains impossible. For instance, pegylation reduces opsonization and interactions with the MPS [101], but also hinders effective extravasation and interstitial diffusion. The desirable properties of liposomes include prolonged retention, enhanced extravasation, and effective interstitial diffusion. However, these disposition processes sometimes compromise one another, meaning that liposome-based designs must consider the integral influence of physicochemical properties on the dispositional processes.

### 4.2. The Gap in Translation from Animal Models to the Clinic

The currently approved liposomes have achieved only a moderate improvement in patient survival compared to the conventional formulations. Therefore, it is vital to determine whether a xenograft tumor model is appropriate for an evaluation of liposomes even though subcutaneous xenogeneic tumor models remain the most popular preclinical model for liposome development. Dramatic differences have been observed between animal tumor models and clinical tumors, and this largely explains the low rate of successful conversion of preclinical efficacy to a clinical survival benefit.

The EPR effect has been broadly observed in xenograft tumor models. It serves as a fundamental principle of nanodrug delivery. However, there is significant heterogeneity in the EPR effect within and between tumor types. It has been reported that different tumor types have different vascular pore sizes and that the maximum pore size changes according to the location of a given type of tumor (i.e., primary vs. metastatic) [83,102]. Moreover, EPR-dependent delivery is often complicated by erratic vascular distribution, high IFP and poor blood perfusion in the tumor [103]. The difference in tumor vascular permeability between the xenograft and a clinical tumor elucidates the low success rate achieved for clinical conversion [104]. Using a xenograft model, it usually takes approximately 2~4 weeks for the tumor to grow to 1 cm (0.5 g) in diameter [105]. In contrast, a clinical diagnosis is only possible after several years. As a result, the tumor vessels in the murine xenograft model are often poorly differentiated and tend to be more permeable than those of a clinical tumor that develops more slowly. Tumor vessel permeability has a close correlation with histologic type and the site of the tumor according to rodent studies [83,102]. A conventional allometric relationship in the clearance of liposomes across mice, rats, dogs, and humans, largely due to inconsistencies in vascular structure and permeability, was not identified in a recent study [106].

### 4.3. The Quantitative Pharmacokinetic–Pharmacodynamic Relationship

Liposomes are designed to augment therapeutic efficacy and reduce toxicity, primarily by influencing the tissue distribution profiles [107]. A quantitative understanding of the multiscale mechanisms of liposomal disposition and influencing factors is critical for the development and regulatory evaluation of liposomal drugs.

The therapeutic efficacy of conventional formulations is generally accurately predicted using systemic PK parameters (e.g., maximum serum/plasma concentration and area under the concentration-time curve). Unlike conventional formulations, the systemic drug exposure of liposomal drugs should not be used to predict efficacy. The only active component of liposomal drugs is the free drug, the active pharmaceutical ingredient (API) released from the liposome. However, the disposition of the API at the target site is usually jointly defined by the local disposition of the API and the drug release kinetics of locally deposited liposomes. Thus, the systemic PK parameters for either intact liposomes or the free drug are not capable of directly predicting the tissue exposure of the free drug and the efficacy and toxicity.

Unfortunately, it remains technically challenging to directly measure the concentration of free APIs in the targeted tissues. The fraction of nuclear doxorubicin to total doxorubicin in a xenograft tumor was reported to be approximately 40~50% and 25% after dosing with Doxil^®^ and dioleoylphosphatidylcholine-pegylated liposomal doxorubicin, respectively, in a study by Laginha et al. [70]. Thus, an integrated PK–PD relationship with regard to the systemic and local kinetics of both the liposomes and free APIs is extremely valuable. It is challenging trying to establish a solid PK–PD relationship for liposomal drugs, but doing so is critical to their development and evaluation.

## 5. The Application of Pharmacokinetic–Pharmacodynamic Modeling and Simulation Systems in Liposomal Drug Delivery

The kinetics of free APIs at the target site should be used to develop the PK–PD relationship for a given liposome. PK–PD modeling and simulation has been used to bypass the challenge of measuring free APIs at the target site. The targeted exposure of free APIs can be predicted by using a PK model that integrates the multiscale dispositions of liposomal drugs, including the systemic and local kinetics of liposomes and the free drugs, as well as drug release. Subsequently, the efficacy can be predicted using a PD model based on the predicted targeted exposure of the free APIs. The developed PK–PD model could be further utilized to identify the critical quality attributes of liposomes in efficacy and to bridge the animal study and clinical study. A series of PK–PD modeling and simulation systems for application to liposomal drugs was identified in this section (Table 2).

### 5.1. Physiologically-Based Pharmacokinetic Modeling and Simulation

As previously discussed, the multiscale mechanisms of liposomal drug disposition are influenced by numerous physiological factors. Zamboni et al. reported that physiological factors (including age, body composition, monocytes, gender, and liver disease status) shape the disposition of liposomal drugs [108,109,110]. Using a “bottom-up” approach, a general whole-body physiologically-based PK (PBPK) model (Figure 2A) and a mechanism-based model for tumor (Figure 2B) can be proposed to capture the detailed disposition of both liposomes and payload in the biological system. Such PBPK models could be used to investigate the influence of drug-associated factors and physiological factors (such as the MPS) on the multiscale mechanisms of liposomal disposition. Currently, PBPK models are widely used in small molecular drug development and regulatory reviews [111,112,113,114,115,116]. PBPK modeling and simulation has recently emerged as a powerful tool for delineating the PK and tissue distribution profiles of liposomal drugs [17,18,117].

#### 5.1.1. Simplified PBPK Model

Harashima et al. developed a hybrid model to correlate the systemic disposition of liposomes with the targeted exposure of the free API [118,119]. A compartmental model was used to describe the systemic PK profiles of encapsulated and released doxorubicin. The tumor, which is the target tissue, was listed as an individual compartment to describe the disposition of encapsulated and released doxorubicin in tumors. The tumor compartment contains capillary, interstitial, and tumor cell sub-compartments and is linked to the systemic compartment by blood flow to the tumor via the capillary sub-compartments. Using this hybrid model, the systemic PK profiles of the liposomes were assessed to predict the tumor distribution profile of free doxorubicin in an attempt to establish the PK–PD relationship. The developed PK–PD framework was then used to assess the influence of the physicochemical and physiological properties of the liposomes on the exposure of the tumor to free doxorubicin and the anti-tumor effect [118]. Liposomal retention in tumors and the local release rate were observed to have a pivotal influence on the anti-tumor effect of liposomal drugs.

The impact of physiological variations on liposomal disposition properties can also be evaluated using PBPK models. Hendriks et al. developed a hybrid PBPK model and assessed the impact of tumor physiology on the cellular and subcellular transport of doxorubicin inside tumors [120]. This work was an example of the use of PBPK modeling and simulation to identify critical physiological factors to improve treatment outcomes and facilitate trans-species or population translation.

Apgar et al. developed a systems model to quantify liposomal RNA disposition and its effect on serum bilirubin [121]. The delivery of RNA to the hepatocytes was quantified by a simplified PBPK model of the kinetics of liposomal RNA degradation, biodistribution, attachment to hepatocytes, endocytic uptake, and intracellular degradation processes in the liver. The effect of the delivered RNA on the regulation of glucuronidation and bilirubin clearance was systematically investigated. Lastly, this PBPK model was applied to bridge preclinical and clinical studies and predict the first human dose.

#### 5.1.2. Whole-Body PBPK Model

Key factors and processes can be integrated using a PBPK model to assess the PK–PD relationship. Kagan et al. developed a dual-layer, whole-body PBPK model to evaluate and describe the disposition of intact liposomes and free amphotericin B (Ambisome^®^) [122]. Drug release kinetics were used to connect the two PBPK layers. Model parameters were optimized to correlate the plasma and tissue concentrations with the time profiles. This was the first whole-body PBPK model to be developed and was capable of simultaneously quantifying the tissue disposition profiles of the encapsulated and free drugs.

### 5.2. PK–PD Modeling System with Spatiotemporal Characterization

The hyperpermeability of blood vessels in a tumor and dysfunction of the lymphatic system enable the accumulation of liposomal drugs in the tumor. This EPR concept is the fundamental principle that is applied to liposomes with respect to the targeted delivery of cytotoxic drugs to solid tumors. The spatial biodistribution of liposomes in tumors is highly heterogeneous [92]. Liposomes in the tumor tissues are mainly located in the perivascular areas and roots of the capillary sprouts, far from the tumor cells. Heterogeneous tumor distribution is believed to explain the moderate improvement in patient survival due to approved liposomal drugs [123]. The heterogeneous distribution of liposomes in tumors is associated with irregular blood vessels in the tumors, elevated IFP, and dense extracellular matrix cell packing. These properties limit the extravasation and penetration of the liposomes. Several system-based models have been proposed to analyze the impact of tumor vasculature and physical properties on heterogeneity in spatial distribution [95,124,125,126].

Macklin et al. developed a mathematical model to describe solid tumor growth and tumor-induced angiogenesis [127]. Based on this model, Frieboes et al. simulated tumor growth and the evolution of the tumor microenvironment over time. They analyzed the influence of the tumor microenvironment properties, particle size, ligand density, and ligand-receptor binding affinity on the spatiotemporal distribution of nanoparticles in tumors [125]. Furthermore, the effects of drug-loading methods, release rates, and diffusivity on treatment efficacy were also assessed [128]. The findings of these studies elucidate the relationship between physiological and physicochemical properties or liposomal disposition and antitumor efficacy.

The impaired blood vessels in tumors expedite the extravasation of nanocarriers into the tumor. However, the increased permeability of the fluid elevates IFP in the tumor, hindering the extravasation of nanoparticles. Thus, normalization of the vasculature can be used to improve extravasation by reducing IFP in the tumor. Jain et al. developed a mathematical model to evaluate the effect of vascular normalization on the delivery of nanomedicines to tumors [95]. A two-dimensional percolation network was constructed with an inlet and an outlet to mimic the tumor microvasculature structure. Blood flow, transvascular fluid flow, interstitial fluid transport, drug transport, and tumor vessel wall permeability were all considered. Vascular normalization improved delivery of the smaller nanoparticles to the tumor, but hindered transport of the larger ones [95]. After electrostatic interactions between the particles and the pores in the vessel wall were taken into account, this model was applied to evaluate the effect of the particle surface charge on the distribution of nanoparticles in the tumor [129]. To further guide particle engineering, this model was also used to investigate the impact of particle size, drug release rate, and targeted binding affinity on intratumor distribution [130,131]. According to their analyses, small particles (≤10 nm) are more effective in tumor distribution and deep penetration. To prevent the fast clearance of small particles and their extravasation to normal tissues, a two-stage approach was proposed to encapsulate the 5-nm secondary particles into a 20-nm primary particle. The release rate is positively related to the antitumor efficacy of nanoparticles, so a relatively high value of release rate is preferable for liposomes. To improve the efficacy, an appropriate targeted binding affinity is required as a relatively high binding rate can trap the drug. The simulation suggested that a binding rate constant of around 1000 (M^−1^s^−1^) was the optimal rate for killing tumor cells.

### 5.3. Combination of In Vitro Study and the PK–PD Modeling System

The systemic disposition profile of liposomal drugs depends on the physicochemical properties of liposomes and the biological system. For instance, the extravasation rate of liposomes is determined not only by the particle size, surface charges, and shapes, but also by the vascular endothelial structures and the permeability of the targeted tissue. Kirtane et al. developed a system-based model to predict the targeted exposure of nanoparticles based on particle and vascular pore sizes [132]. They concluded that a particle size that was universally optimal for all types of tumors could not be identified. Optimal particle properties were found to be contingent on the type and features of the tumor. The high variability of experimental animals often confounds the identification of influencial particle parameters on each process of particle disposition. An in vitro assay under well-controlled biological conditions is a more desirable tool than the in vivo study for identifying the effect of particle properties on particle disposition. However, how to translate the in vitro assay result to in vivo liposome disposition prosperities remains a major obstacle. In vitro—in vivo correlation (IVIVC) is defined by the FDA as “a predictive mathematical model that is used to describe the relationship between the in vitro property of a dosage form and a relevant in vivo response” [133]. Once IVIVC has been established and validated, the results of in vitro studies can be used to predict drug performance in vivo. The use of a combination of well-controlled high-throughput in vitro assays and IVIVC approaches between particle properties and the in vivo disposition (e.g., clearance, extravasation) of the liposomes holds promise in liposome optimization. Moreover, PK–PD modeling system could integrate IVIVC to predict the targeted exposure of API and then the efficacy of the liposome based on the in vitro measurable particle properties.

Recently, Mayer et. al. developed an IVIVC approach to predict liposome in vivo release [134]. Multilamellar vesicles were used as “acceptors” to mimic the in vivo membrane and predict drug release in vivo. This in vitro release testing method accurately predicted the in vivo release characteristics of liposomal drugs compared to the commonly used dialysis membrane method. The lipid membrane pool in the circulation and tissues helps to explain the poor correlation between in vitro and in vivo drug release.

In a similar study, a two-stage reverse dialysis method (i.e., dialysis in pH 7.4 HEPES buffer solution [stage 1] and dialysis in 1% TX100 solution in HEPES buffer solution [stage 2]) was developed to mimic the release of the drug into the blood and targeted tissue [135]. Premature drug release can also be induced by plasma protein adsorption and the subsequent identification of liposomes by the MPS. In an attempt to predict systemic clearance, Crielaard et al. characterized liposome–protein interactions and liposome aggregations using surface plasmon resonance and single-molecule tracking fluorescence microscopy [136]. Aggregation was not observed, and clearance of the liposomes was seen to positively relate to interactions between the liposomes and proteins, suggesting that liposome–protein interactions can be used to predict the systemic clearance of liposomes.

The developed IVIVC could be incorporated into a PBPK model to quantify the effect of liposomal properties on the multiscale mechanisms of liposomal disposition. Based on a meta-analysis of data in the literature, an IVIVC was developed in our unpublished study between liposomal particle size, surface charge, and pegylation and systemic clearance and tumor disposition [137]. These relationships were then incorporated into a PBPK model of liposomal doxorubicin to identify the most sensitive liposomal property characteristics in relation to high efficacy and low toxicity. Importantly, this integrated modeling framework has the capacity to identify critical or trivial physicochemical properties, which would potentially benefit the regulatory reviews that are conducted to evaluate generic liposomal drugs.

## 6. Perspective and Future Direction

Liposomal formulations have been developed to improve the therapeutic index of encapsulated drugs by altering the balance of on- and off-target distribution. The use of PK–PD modeling and simulation systems is considered a powerful means of identifying system- and drug-associated factors that influence the multiscale mechanisms of liposomal disposition. Several challenges need to be addressed in order to further improve PK–PD models to enhance the development and regulation of liposomal drugs.

Firstly, it is important to note that the therapeutic effects of liposomal drugs are elicited by the free payload at the site of action. However, reliable quantification-based approaches to measure free payloads in low abundance at the targeted sites are still lacking, particularly in clinical situations. Although PK–PD modeling and simulation systems are able to predict the targeted exposure of free payloads, experimental measurement of the free drug exposure at the targeted sites is required for additional model verification.

Secondly, the key advantages of PK–PD modeling and simulation include the ability to integrate the multiscale disposition mechanisms of liposomal disposition and elucidate the quantitative relationships between the physicochemical properties of liposomes and the systemic and target disposition behavior (IVIVC). Unfortunately, only the IVIVC between drug release and liposome clearance has been established. IVIVCs between other liposomal properties and other liposome disposition processes (e.g., penetration, extravasation, and endocytosis) have not been reported in literature. Moreover, the available IVIVC is highly formulation- or animal model-dependent and needs to be verified before scaling to other scenarios.

Thirdly, the developed PK–PD modeling and simulation systems or PBPK models take into account many physiological factors, but many of these are highly variable across populations and some of them are even influenced by the treatment of liposomal drugs. This creates additional sources of variability in model prediction and simulation. When using PK–PD modeling and simulation models for clinical translation, it is important to consider variations in the sociodemographic characteristics of the patients and to evaluate whether it would result in clinically meaningful changes in response to liposomal drug treatment.

In conclusion, PK–PD modeling and simulation systems provide an integrated and upgradable approach to guide the development and regulation of liposomal drugs. The present review summarized the physiological factors that affect the targeted delivery of liposomal drugs, challenges that influence the development and regulation of liposomal drugs, and the application of PK–PD modeling and simulation systems to address these challenges.

## Figures and Tables

**Figure 1 pharmaceutics-11-00110-f001:**
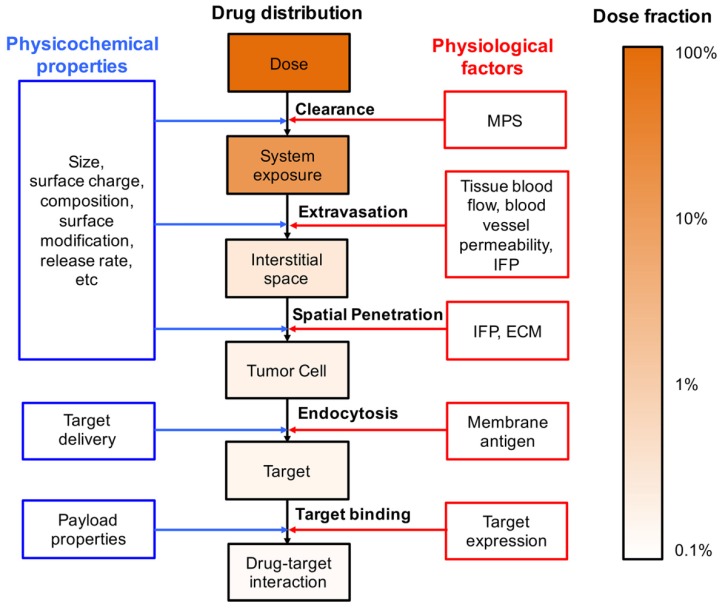
Multiscale physiological barriers to the targeted distribution of liposomal drugs. The dose fraction change was obtained from references [70,71]. MPS—mononuclear phagocyte system; IFP—interstitial fluid pressure; ECM—extracellular matrix.

**Figure 2 pharmaceutics-11-00110-f002:**
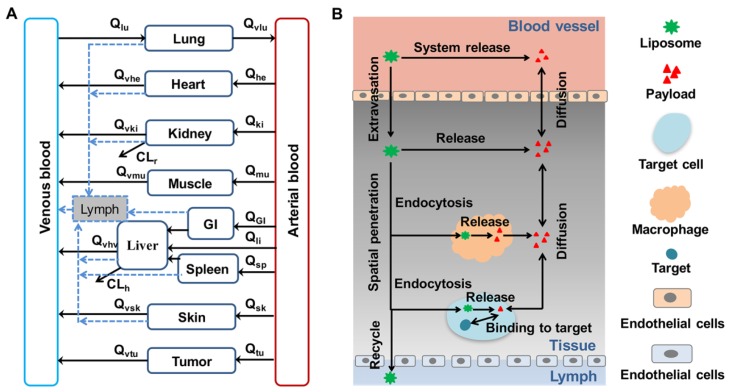
A general PBPK model diagram for liposomal drugs. (**A**) A generic dual-layer PBPK model. The black solid line represents the blood flow and the blue dashed line depicts the lymph flow. (**B**) A tumor tissue compartment model for liposomes. The distribution from blood to drug target involves liposome extravasation, diffusion of the released payload between blood and interstitial fluid, direct or tissue-associated macrophage-mediated release of the payload, tumor cell endocytosis of liposomes, intracellular release of the payload in tumor cells, target binding, and recycling of intact liposomes through lymph back to the blood. Q—blood flow; CL_r_—renal clearance; CL_h_—hepatic clearance.

**Table 1 pharmaceutics-11-00110-t001:** Summary of the approved liposomal drugs for use in humans by the U.S. Food and Drug Administration (FDA) and the European Medicines Agency (EMA).

Product	Active Drug	Therapeutic Area	Year of Approval	PK–PD Profiles
DaunoXome^®^	Daunorubicin	Cancer	1996	Prolonged retention, increased target distribution, equivalent efficacy, and reduced toxicity.
DepoCyt^®^	Cytarabine	Cancer	1999	Prolonged tumor exposure to cytotoxic concentration, increased response rate, and reduced toxicity.
Doxil^®^	Doxorubicin	Cancer	1995	Prolonged retention, increased target distribution, equivalent efficacy, and reduced toxicity.
Lipodox^®^	Doxorubicin	Cancer	2012	Prolonged retention, increased target distribution, equivalent efficacy, and reduced toxicity.
Myocet^®^	Doxorubicin	Cancer	2000	Prolonged retention, increased target distribution, equivalent efficacy, and reduced toxicity.
Marqibo^®^	Vincristine	Cancer	2012	Prolonged retention, increased target distribution, superior efficacy, and reduced toxicity.
Vyxeos^®^	Daunorubicin-cytarabine (1:5)	Cancer	2017	Increased targeted exposure of daunorubicin and cytarabine in a fixed ratio, superior efficacy, and comparable toxicity.
Onivyde^TM^	Irinotecan	Cancer	2015	Prolonged retention and increased exposure of the bioactive metabolite of irinotecan (SN-38), superior efficacy, and reduced toxicity.
Mepact^®^	Mifamurtide	Cancer	2009	Prolonged and increased retention in target, superior efficacy, and reduced toxicity.
Ambisome^®^	Amphotericin B	Infection	1997	Releases the drug only when the liposome binds to the fungus and significant reduction in toxicity.
Arikayce^®^	Amikacin	Infection	2018	Increased target distribution, superior efficacy, and comparable toxicity.
Inflexal^®^ V	Flu vaccine	Vaccine	1997	Superior immune response.
Epaxal^®^	Hepatitis A vaccine (synthetic lipids, influenza proteins, hepatitis A antigen)	Vaccine	1994	Higher tolerability and reduced toxicity.
DepoDur^TM^	Morphine	Analgesics	2004	Prolonged retention, reduced peak concentration, superior efficacy, and reduced toxicity.
Exparel^®^	Bupivacaine	Analgesics	2011	Prolonged retention, reduced peak concentration, superior efficacy, and reduced toxicity.
Visudyne^®^	Verteporfin	Photodynamic therapy	2000	Equivalent clearance, slightly increased tissue distribution, and reduced toxicity.

**Table 2 pharmaceutics-11-00110-t002:** Pharmacokinetic–Pharmacodynamic (PK–PD) modeling systems in liposomal drug delivery.

Model Approach	Mathematical Model	Mathematical Model of the Rest of the Body	Drugs	Notes	Ref
Simplified physiologically-based pharmacokinetic (PBPK) model Simplified PBPK model Simplified PBPK model	Tumor was divided into capillary, interstitial and tumor cell sub-compartments.	1-compartment PK model for liposome and 2-compartment PK model for drug.	Doxorubicin	Liposomal retention in tumors and the local release rate were identified to play pivotal roles in antitumor efficacy.	[118,119]
	Tumor was divided into capillary, interstitial, tumor cell and nucleus sub-compartments.	1-compartment PK model for liposome and2-compartment PK model for drug.	Doxorubicin	The detailed drug transport into and out of the cell, drug-target association and dissociation, and liposome uptake and release in tumor cells were described.	[120]
	Liver hepatocyte compartment with endosomal and cytoplastic compartments.	Plasma compartment.	hUGT1A1-modRNA	Endocytosis, release and transcription processes were described. After translated to humans, this model was used to estimate the first-in-human dose.	[121]
Whole-body PBPK model	/	Plasma and tissue (liver, spleen, kidneys, gut, lungs, heart and others) compartments.	Amphotericin B	The first whole-body PBPK model described the disposition of both liposome and drug simultaneously.	[122]
Model with spatiotemporal characterization Model with spatiotemporal characterization	The combination of tumor growth, angiogenesis, oxygen transport, nanoparticle transport and antitumor effect models.	/	/	The physiological properties of the tumor were considered. The model described the interactions between tumor progression and liposome disposition.	[125,127,128]
	Tumor vascular network and nanoparticle transport model.	/	/	Tumor blood vessel properties were simulated in this model. The interactions between the liposome and blood vessel were simulated to optimize the particle properties.	[129,130,131]
Model with in vitro—in vivo correlation (IVIVC)	Using in vitro studies to determine the model parameters and replace in vivo study in liposome optimization study.	Plasma and tissues compartments.	Doxorubicin	Liposome property–disposition relationships were established to facilitate liposome optimization.	[137]

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
