# Peer review of "Pharmacokinetics and Pharmacodynamics Modeling and Simulation Systems to Support the Development and Regulation of Liposomal Drugs"

_pharmaceutics, 2019, doi:10.3390/pharmaceutics11030110_

Round 1

Reviewer 1 Report

He et al have written a comprehensive review about the various clinically-approved liposome-encapsulated drugs, the mechanisms of their disposition in the body and the critical factors that predict or define their efficacy. Importantly, the authors review the different models and simulations reported in the literature that can potentially form a basis for improved models for predicting the in vivo disposition of liposome-encapsulated drugs which in turn could support the development or regulatory reviews of such drugs. Although the review is fairly thorough, addressing the following concerns will improve the manuscript:

Major concern: the section 'The application of systems pharmacokinetic-pharmacodynamic modeling and simulation to liposomal drug delivery' is not structured well. The structure should be either improved or a small paragraph describing the summary of different models and simulation described in this section should be included along with how they can be combined to build a better model.

Minor concerns:

1) No comments about the clinical efficacy/results of DepoCyt are included. The authors should include these. 

2) In line line 189, it is not clear how DepoDur 'expedites' the release of drug into the epidural space. The authors should clarify.

3) In lines 198-199, the authors should clarify what they mean by tolerance to the increased dose.

4) In lines 212-214, the authors claim that liposomes are designed to increase the permeability - the authors should explain the mechanism behind this.

5) In Figure 1, the authors should use 'spatial penetration' instead of 'penetration'.

6) In line 260, 'convention' should be replaced by 'convection'.

7) In line 270, 'reduction in the hypertension of the extracellular matrix' should be replaced by 'modification or degradation of extracellular matrix'.

8) In line 282, the authors state 'liposomes are sorted into endosomes and lysosomes'. The authors should rephrase this phrase since liposomes initially enter endosomes, which mature into late endosomes and then into lysosomes.

9) A reference should be included for lines 311-312.

10) The authors should clarify the species they are referring to in lines 324-325.

11) Lines 327-330 should be corrected since, SCID, Nude and Rag KO are most commonly used for xenograft studies and the MPS system in these immunodeficient mice models is generally considered to be unperturbed (these mice mainly suffer from adaptive immune cell deficiencies).

12) In line 337, 'maximum concentration' should be replaced by 'maximum serum/plasma concentration'.

13) Lines 342-343 and 359-361 are not clear. Please explain better.

14) The authors should add 2-3 sentences to lines 372-373 to explain the 'tumor compartment' mentioned here.

15) In Figure 2, Endothelia cells should be replaced with Endothelial cells. 

16) Figure 2 is not cited in the text. 

17) Lines 420-421 should be rephrased - this heading is not clear.

18) Lines 434-436 should be rephrased or explained better.

19) In line 455, it is not clear what is the 'current study'.

20) The authors should clarify what they mean by 'far removed from the tumor cells' in line 468.

21) The statement in lines 495-497 should be expanded to list the favorable properties of liposomes or effects of different parameters.

22) Lines 509-510 and 514-516 are not clear and should be explained better or rephrased.

Author Response

Responses to Reviewers: pharmaceutics-447189

We sincerely appreciated both reviewers for these great suggestions and comments. In this response, bold and italic were newly added or modified contents, respectively. The referred line numbers in this response were listed according to the clean version.

Major concern: the section 'The application of systems pharmacokinetic-pharmacodynamic modeling and simulation to liposomal drug delivery' is not structured well. The structure should be either improved or a small paragraph describing the summary of different models and simulation described in this section should be included along with how they can be combined to build a better model.

Response: Thanks for your comments and suggestion. We have improved the structure of this section by dividing it into three parts:

(1) Physiologically based pharmacokinetic modeling and simulation.

(2) System PK/PD model with spatiotemporal characterization.

(3) Combination of in vitro study and system PK/PD model.

Minor concerns:

1) No comments about the clinical efficacy/results of DepoCyt are included. The authors should include these. 

Response: We have added the clinical efficacy in lines 137-140: ‘The slow and continuous release profile of liposome resulted in extended (up to 14 days) and enhanced tumor exposure to cytarabine, thus increasing the response rate when compared with the free cytarabine treatment [45-48].’

2) In line line 189, it is not clear how DepoDur 'expedites' the release of drug into the epidural space. The authors should clarify.

Response: We have modified the description in lines 189-190: ‘Morphine is encapsulated in lipid foam using DepoFoam™ technology [65] to allow its extended release into the epidural space.’

3) In lines 198-199, the authors should clarify what they mean by tolerance to the increased dose.

Response: We have modified the sentences as in lines 196-200: ‘Compared to the plain formulation, the peak concentrations of bupivacaine for liposomal bupivacaine were comparable (0.87 vs. 0.83 μg/ml) at a 4-fold higher dose, and the time to the maximal concentration was 7-fold greater [67], resulting in lower toxicity. The higher tolerance dose and the delayed elimination of bupivacaine in the liposome formulation produce prolonged analgesia.

4) In lines 212-214, the authors claim that liposomes are designed to increase the permeability - the authors should explain the mechanism behind this.

Response: We have deleted ‘increase the permeability’ in lines 210-217: ‘The pharmacological effects of chemical compounds are usually limited by their poor physicochemical and PK properties, such as low solubility, suboptimal biodistribution, and rapid clearance. Liposomes are designed to overcome these challenges through the encapsulation of chemical compounds, thereby increasing drug solubility, reducing systemic clearance, and improving the drug distribution to the target tissue.

5) In Figure 1, the authors should use 'spatial penetration' instead of 'penetration'.

Response: Thanks for your comment. We have corrected it in Figure 1.

6) In line 260, 'convention' should be replaced by 'convection'.

Response: Thanks for your comment. We have corrected it in line 261.

7) In line 270, 'reduction in the hypertension of the extracellular matrix' should be replaced by 'modification or degradation of extracellular matrix'.

Response: We have changed 'reduction in the hypertension of the extracellular matrix' to ‘degradation of extracellular matrix’ in line 270-271 as your suggestion.

8) In line 282, the authors state 'liposomes are sorted into endosomes and lysosomes'. The authors should rephrase this phrase since liposomes initially enter endosomes, which mature into late endosomes and then into lysosomes.

Response: This phrase has been changed in lines 282-285: ‘Once taken up by the tumor cells via endocytosis, the liposomes in endosome are either recycled to the cell surface or degraded by acidic conditions or the residing enzymes after the endosome matured into late endosome and then fused with lysosome [98,99].’

9) A reference should be included for lines 311-312.

Response: Reference (83, 105) was added in line 320.

10) The authors should clarify the species they are referring to in lines 324-325.

Response: We have added the species in line 328. “Tumor vessel permeability has a close correlation with histologic type and the site of the tumor according to rodent studies.

11) Lines 327-330 should be corrected since, SCID, Nude and Rag KO are most commonly used for xenograft studies and the MPS system in these immunodeficient mice models is generally considered to be unperturbed (these mice mainly suffer from adaptive immune cell deficiencies).

Response: We agree with the reviewer and deleted these statements.

12) In line 337, 'maximum concentration' should be replaced by 'maximum serum/plasma concentration'.

Response: We have added “serum/plasma” in line 337: ‘The therapeutic efficacy of conventional formulations is generally accurately predicted using systemic PK parameters (e.g., maximum serum/plasma concentration and area under the concentration-time curve).’

13) Lines 342-343 and 359-361 are not clear. Please explain better.

Response: Lines 342-343 were rephrased as lines 338-344: ‘Unlike conventional formulations, the systemic drug exposure of liposomal drugs should not be used to predict efficacy. The only active component of liposomal drugs is the released free drug, the active pharmaceutical ingredients (API) released from the liposome. The disposition of API at the target site is usually jointly defined by the local disposition of API and the drug release kinetics of locally deposited liposomes. Thus, the systemic PK parameters for either intact liposomes or free drug are not capable of directly predicting the tissue exposure of free drug and the efficacy and toxicity.’

Lines 342-343 were rephrased as lines 352-359: ‘The kinetics of free APIs at the targeted site can be used to develop the PK/PD relationship for a given liposome. Using a PK model that integrates the multiscale dispositions of liposomal drugs, including the systemic and local kinetics of liposomes and free drugs, as well as drug release, the target exposure of free APIs can be predicted. Subsequently, the efficacy can be predicted using a PD model based on the predicted target exposure of the free APIs. The developed PK/PD model could be further utilized to identify the critical quality attributes of liposomes in efficacy and to bridge the animal study and clinical study. A series of PK/PD modeling and simulation systems for application to liposomal drugs was identified in this section.’

14) The authors should add 2-3 sentences to lines 372-373 to explain the 'tumor compartment' mentioned here.

Response: We explained in lines 386-390. ‘Tumor, the target tissue, was listed as an individual compartment to describe the disposition of encapsulated and released doxorubicin in the tumors. The tumor compartment contains a capillary, interstitial, and tumor cell sub-compartments and was linked to the systemic compartment by blood flow to the tumor via the capillary sub-compartments.

15) In Figure 2, Endothelia cells should be replaced with Endothelial cells. 

Response: Thanks. We have modified it.

16) Figure 2 is not cited in the text. 

Response: We have cited Figure 2 in line 367-369. “Using a “bottom-up” approach, a general whole-body physiologically based PK (PBPK) models (Figure 2A) and a mechanism-based model for tumor (Figure 2B) can be proposed to capture detailed disposition of both liposome and payload in the biological system with mathematical equations.

17) Lines 420-421 should be rephrased - this heading is not clear.

Response: This heading was rephrased in line 458:Combination of in vitro study and system PK/PD model’.

18) Lines 434-436 should be rephrased or explained better.

Response: We have added more explanation in 459-478: ‘The systemic disposition profile of liposomal drugs depends on the physicochemical properties of liposomes and also the biological system. For instance, the extravasation rate of liposomes is determined not only by the particle size, surface charges, and shapes, but also by the vascular endothelial structures and permeability of the targeted tissue. Kirtane et al. developed a system-based model to predict the targeted exposure of nanoparticles based on particle and vascular pore sizes [132]. They concluded that a particle size that was universally optimal for all types of tumors could not be identified. Optimal particle properties were found to be contingent on the type and features of the tumor. The high variability of experimental animals often confounds the identification of influencial particle parameters on each process of particle disposition. Compared with in vivo study, in vitro assay under well-controlled biological conditions is a more desirable tool for identifying the effect of particle properties on particle disposition. However, how to translate the in vitro assay result to in vivo liposome disposition prosperities remains an major obstacle. In vitro-in vivo correlation (IVIVC) is defined by the FDA as “a predictive mathematical model that is used to describe the relationship between the in vitro property of a dosage form and a relevant in vivo response” [133]. Once IVIVC has been established and validated, the results of in vitro studies can be used to predict drug performance in vivo. The use of a combination of well-controlled high-throughput in vitro assays and IVIVC approaches between particle properties and in vivo disposition (e.g., clearance, extravasation) of the liposomes holds promise in liposome optimization. Moreover, systems PK/PD models could integrate IVIVC to predict the target exposure of API and then the efficacy of liposome based on the in vitro measurable particle properties.

19) In line 455, it is not clear what is the 'current study'.

Response: we have changed it to ‘our unpublished study’ in line 497.

20) The authors should clarify what they mean by 'far removed from the tumor cells' in line 468.

Response: We have made the correction in line 422: “far from the tumor cells”.

21) The statement in lines 495-497 should be expanded to list the favorable properties of liposomes or effects of different parameters.

Response: More explanation was added in lines 450-457: ‘According to their analyses, particles of smaller size ( 10 nm) are more effective in tumor distribution and deep penetration. To prevent the fast clearance of small particles and their extravasation to normal tissues, a two-stage approach was proposed to encapsulate the 5-nm secondary particles into a 20-nm primary particle. The release rate is positively related to the antitumor efficacy of nanoparticles, so a relatively high value of release rate is preferable for liposome. To improve efficacy, an appropriate targeted binding affinity is required as relatively high binding rate can trap the drug. The simulation suggested that a binding rate constant around 1000 (M-1s-1) was the optimal rate for tumor cells killing.’

22) Lines 509-510 and 514-516 are not clear and should be explained better or rephrased.

Response: Lines 509-510 was rephrased as lines 514-516: ‘Although PK/PD modeling and simulation systems are able to predict the targeted exposure of free payloads, experimental measurement of free drug exposure at the targeted sites is required for additional model verification.

Lines 514-516 was rephrased as lines 517-524: ‘Secondly, the key advantages of PK/PD modeling and simulation include the ability to integrate the multiscale disposition mechanisms of liposomal disposition and elucidate the quantitative relationships between the physicochemical properties of liposomes and the systemic and target disposition behavior (IVIVC). Unfortunately, only the IVIVC between drug release and liposome clearance has been established. IVIVCs between other liposomal properties and other liposome disposition processes (e.g., penetration, extravasation, and endocytosis) have not been reported in literature. Moreover, the available IVIVC is highly formulation- or animal model-dependent and needs to be verified before scaling to other scenarios.

Reviewer 2 Report

This manuscript reports the review on Systems pharmacokinetics and pharmacodynamics

modeling and simulation to support the development and regulation of liposomal drugs. Overall this manuscript is reasonably organized and well-written. There are some suggestions or questions to be improved.

1. Authors try to use Table 1 to summarize the approved liposomal drugs for use in humans by either the FDA or the EMA. However, it is suggested that the content from Table 1 should include the pharmacokinetic aspects of liposomal drugs, not the indication.

2. Third section of this review manuscript tries to describe the application of systems pharmacokinetic-pharmacodynamic modeling and simulation to liposomal drug delivery. The reviewer think that that section is quite important and interesting to the reader of this paper. A new well-organized Table should be provided for the summary of PK/PD approaches in this review.

3. Please indicate a proper reference to support dose fraction change, shown in Figure 1.

4. Figure 2 was provided without any description in the manuscript. Please add a detail description and discussion to be related in Figure 2 somewhere.

5. There are so many typos and errors. Please revise carefully. For examples, Table 1 has so may mistakes for capital letter (daunorubicin, amikacin, influenza , infection…..)

6. Why does DounoXome has bold style in Table 1?

7. P6 line 307, please use abbreviation only for EPR.  

Author Response

Responses to Reviewers: pharmaceutics-447189

We sincerely appreciated both reviewers for these great suggestions and comments. In this response, bold and italic were newly added or modified contents, respectively. The referred line numbers in this response were listed according to the clean version.

Reviewer #1

1. Authors try to use Table 1 to summarize the approved liposomal drugs for use in humans by either the FDA or the EMA. However, it is suggested that the content from Table 1 should include the pharmacokinetic aspects of liposomal drugs, not the indication.

Response: thanks for your suggestion. The PK and PD aspects of liposomal drugs were included in Table1.

2. Third section of this review manuscript tries to describe the application of systems pharmacokinetic-pharmacodynamic modeling and simulation to liposomal drug delivery. The reviewer think that that section is quite important and interesting to the reader of this paper. A new well-organized Table should be provided for the summary of PK/PD approaches in this review.

Response: we have summarized the PK/PD approaches in Table 2.

3. Please indicate a proper reference to support dose fraction change, shown in Figure 1.

Response: we have added the references (70, 71) in line 220.

4. Figure 2 was provided without any description in the manuscript. Please add a detail description and discussion to be related in Figure 2 somewhere.

Response: We described Figure 2 in lines 367-369. “Using a “bottom-up” approach, a general whole-body physiologically based PK (PBPK) models (Figure 2A) and a mechanism-based model for tumor (Figure 2B) can be proposed to capture detailed disposition of both liposome and payload in the biological system with mathematical equations.

5. There are so many typos and errors. Please revise carefully. For examples, Table 1 has so may mistakes for capital letter (daunorubicin, amikacin, influenza , infection…..)

Response: Sorry for these mistakes. We have checked and corrected them.

6. Why does DounoXome has bold style in Table 1?

Response: Sorry for this format error. We have corrected it.

7. P6 line 307, please use abbreviation only for EPR.  

Response: We have corrected it.

Round 2

Reviewer 2 Report

The revised manuscipt tries to reflect the reviewer's suggestions and comments properly.